

# RdRp mutations are associated with SARS-CoV-2 genome evolution

Doğa Eskier[1,2,*], Gökhan Karakülah[1,2,*], Aslı Suner[3] and Yavuz Oktay[1,2,4]

[1] Izmir Biomedicine and Genome Center (IBG), Izmir, Turkey
[2] Izmir International Biomedicine and Genome Institute, Dokuz Eylül University, Izmir, Turkey
[3] Department of Biostatistics and Medical Informatics, Faculty of Medicine, Ege University, Izmir, Turkey
[4] Faculty of Medicine, Department of Medical Biology, Dokuz Eylül University, Izmir, Turkey
* These authors contributed equally to this work.

## ABSTRACT

COVID-19, caused by the novel SARS-CoV-2 virus, started in China in late 2019, and soon became a global pandemic. With the help of thousands of viral genome sequences that have been accumulating, it has become possible to track the evolution of the viral genome over time as it spread across the world. An important question that still needs to be answered is whether any of the common mutations affect the viral properties, and therefore the disease characteristics. Therefore, we sought to understand the effects of mutations in RNA-dependent RNA polymerase (RdRp), particularly the common 14408C>T mutation, on mutation rate and viral spread. By focusing on mutations in the slowly evolving M or E genes, we aimed to minimize the effects of selective pressure. Our results indicate that 14408C>T mutation increases the mutation rate, while the third-most common RdRp mutation, 15324C>T, has the opposite effect. It is possible that 14408C>T mutation may have contributed to the dominance of its co-mutations in Europe and elsewhere.

## INTRODUCTION

SARS-CoV-2 is a novel betacoronavirus originally identified in December 2019, and given the official name on 11 February 2020. It is responsible for the ongoing COVID-19 pandemic, with the earliest known patients located potentially as early as November 2019, in the Hubei province of China. Human to human transmission of the virus was confirmed on 20 January 2020 (*Chan et al., 2020*), with six deaths and 282 confirmed cases on 21 January 2020, which, as of 13 May 2020, had respectively increased to over 283,000 deaths and 4.09 million cases, with a projected mortality rate of <7%, and an R0 number estimation of 1.4–3.8 (*Riou & Althaus, 2020*). Due to its high transmission rate, and worldwide distribution of known cases, SARS-CoV-2 is a high priority for medical research despite the low mortality rate. Furthermore, new COVID-19 symptoms continue to be discovered even in recovering patients (*Wilson, Katlariwala & Low, 2020*; *Li et al., 2020*), making it difficult to fully understand the global impact of the disease. To date, there is no known targeted treatment or vaccine against SARS-CoV-2.

Corresponding author
Yavuz Oktay, yavuz.oktay@ibg.edu.tr

The current reference genome for SARS-CoV-2, NC_045512.2, was sequenced in Shanghai, China, and submitted on 5 January 2020, replacing the previous reference sequence on 17 January 2020. SARS-CoV-2 has a 29903-nucleotide long single stranded sense RNA genome, which codes for 12 peptides, including two closely related polyproteins, Orf1a and Orf1ab, which are further cleaved into 26 mature peptides, and the main structural proteins, such as the surface glycoprotein, nucleocapsid phosphoprotein, membrane glycoprotein, and envelope glycoprotein (genes S, N, M and E, respectively) ("Severe acute Respiratory Syndrome Coronavirus 2 Isolate Wuhan-Hu-1, Complete Genome," 2020, http://www.ncbi.nlm.nih.gov/nuccore/NC_045512.2). The primary binding target of SARS-CoV-2 surface glycoprotein for entry into the human cell is the ACE2 protein (*Tian et al., 2020*), localized in the cell membrane in a number of tissues (*Hikmet et al., 2020*). It is predicted that the virus is zoonotic in origin, and mutations in the surface glycoprotein structure enabled transmission to human hosts. As a result of the origins of the disease, and due to the targeted nature of vaccine and drug discovery efforts, identifying the replication-related mutation rate and the global mutatome of SARS-CoV-2 is crucial to efforts in combating the disease. Despite having proof-reading capability, analyses of SARS-CoV-2 genomes indicated nucleotide substitution rates comparable to other RNA viruses that lack such capability (*Zhang & Holmes, 2020*). It is difficult to pinpoint the underlying causes without functional studies; however, one plausible explanation would be reduced fidelity of the main RNA polymerase, namely RNA-dependent RNA polymerase (RdRp, also known as nsp12), due to mutations.

SARS-CoV-2 does not depend on host polymerases to replicate its genome, instead using the RdRp and associated proteins (i.e., nsp7, nsp8 and nsp14, the latter an exonuclease with error correction capabilities), all of which are encoded by its own genome (*Subissi et al., 2014*; *Ma et al., 2015*). Among the SARS-CoV-2 isolates from all over the world, several widespread mutations on the RdRp coding region of the Orf1ab polyprotein gene have been identified. Key among them is the 14408C>T transition, identified in over 7,000 isolates across multiple continents. In an early analysis of 137 SARS-CoV-2 genomes from North America and Europe, *Pachetti et al. (2020)* suggested that this particular mutation is associated with higher number of mutations, through a mechanism not yet fully known. The proline to leucine substitution (P323L) caused by the 14408C>T mutation has been suggested to rigidify the RdRp protein structure, which may exert its effects through altered interaction with other components of the replication/transcription machinery or with the RNA template, thereby resulting in an altered mutation rate (*Pachetti et al., 2020*; *Begum et al., 2020*). However, further studies are needed to test these hypotheses.

Although the higher number of mutations in genomes with RdRp 14408C>T mutation was suggested to be caused by lower fidelity of the mutant enzyme, it could also be due to many other epidemiological factors, such as worse containment of the mutant strain compared to others, more frequent introduction of the mutant strain into high-risk and/or higher-transmitting populations by chance, etc. Importantly, natural selection acting on different levels and ways in different environments could potentially lead to faulty estimates of any change in the replicational mutation rate. Therefore, in order to minimize

the effects of natural selection, we focused our attention on parts of the genome that may be under lower selective pressure.

A recent study by *Dilucca et al. (2020)* showed that different SARS-CoV-2 genes are under varying levels of selective pressure, with M and E integral proteins being subject to relatively low natural selection and a low, non-selective mutation rate, largely as a result of accumulation of replication errors. On the other hand, key proteins for virulence and transmissibility, such as the S protein, seem to be under high selective pressure, possibly as a result of novel host adaptation (*Dilucca et al., 2020*). To identify how the RdRp mutations affect the mutation rate of the SARS-CoV-2 genome, we examined the relationships of the RdRp mutations with those found in M or E proteins (hereafter referred to as MoE), in terms of both time and location. In particular, we focused on the 10 most common mutations in the RdRp region, with the goal of identifying whether each variant is associated with increased or decreased non-selective mutation rates, and whether the geographical distribution of these variants might suggest the presence of multiple forms of virus with various mutability across the globe.

## MATERIALS AND METHODS

### Genome sequence filtering, retrieval and preprocessing

SARS-CoV-2 isolate genome sequences were obtained from the GISAID EpiCoV database ("GISAID Initiative", https://www.epicov.org/epi3/frontend#272e13). The genomes were filtered for those obtained from human hosts, a sequence length of at least 29,000 nucleotides, and high coverage (<1% undefined nucleotides, <0.05% mutations not seen in any other isolate, and no indel mutations that were not verified by the submitter). The filters resulted in a total of 11,901 remaining genomes (as of 5 May 2020). The genomes were aligned against the reference genome sequence obtained from the NCBI Nucleotide database in the FASTA format, under locus ID NC_045512.2 (https://www.ncbi.nlm.nih.gov/nuccore/NC_045512.2), after nonstandard unresolved base calls (sequence characters which are not A, C, G, T, N, or -) were changed into the standard unresolved sequence character N via the Unix *sed* command. The alignment was performed with the MAFFT multiple sequence alignment program, using the "--auto --keeplength --addfragments isolate_genomes.fa reference_genome.fa > alignment.fa" parameters. The sites differing from the reference sequence were extracted using snp-sites (https://github.com/sanger-pathogens/snp-sites), with the "-v -o variants.vcf alignment.fa" options. The resulting VCF file was modified for compatibility with the following steps using text editing and bcftools (http://www.htslib.org/download/), replacing the first column, indicating reference sequence name, with NC_045512v2, and separating different variants at the same nucleotide to individual lines, using the VCF processing guide available in the ANNOVAR documentation (https://doc-openbio.readthedocs.io/projects/annovar/en/latest/articles/VCF/). The final VCF file was converted into an avinput file, using convert2annovar.pl found under ANNOVAR, with the parameters "-format vcf4old variants.vcf > variants.avinput". The custom ANNOVAR gene annotations for SARS-CoV-2 were obtained from ANNOVAR resources, decompressed, and placed in the sarscov2db directory. The variants were then annotated

in terms of their relationships to gene loci and products, using the table_annovar.pl function of ANNOVAR, with the parameters "-buildver NC_045512v2 variants.avinput sarscov2db/ -protocol avGene -operation g".

Following the alignment and annotation, the 5′ untranslated region of the genome (bases 1–265) and the 100 nucleotides at the 3′ end were removed from analysis due to lack of quality sequencing in a majority of isolates. To ensure a vigorous examination of the association of both time and location and the mutations, we have further filtered out isolate genomes without well-defined time of sequencing metadata (year–month–day), and an undefined geographical location, for a final count of 11,208 genomes. A total of 71 of these genomes were sequenced in Africa, 859 were sequenced in Asia, 5,769 in Europe, 3,370 in North America, 1,021 in Oceania, and 118 in South America.

### Statistical analysis

Descriptive statistics for continuous variable days were calculated with mean, standard deviation, median and interquartile range. Shapiro–Wilk test was used to check the normality assumption of the continuous variable. In cases of non-normally distributed data, the Wilcoxon rank-sum (Mann–Whitney $U$) test was performed to determine whether the difference between the two MoE groups was statistically significant. The Fisher's exact test and the Pearson chi-square test were used for the analysis of categorical variables. The univariate logistic regression method was utilized to assess the mutations associated with MoE in single variables, and then multiple logistic regression method was performed. The final multiple logistic regression model was executed with the backward stepwise method. A $p$-value of less than 0.05 was considered statistically significant. All statistical analyses were performed using IBM SPSS version 25.0 (Chicago, IL, USA).

## RESULTS AND DISCUSSION

### Mutation profile of SARS-CoV-2 genome as of 5 May 2020

After the low quality filters were applied, 5,658 nucleotides, making up 18.9% of the SARS-CoV-2 genome, were found to carry a mutation in at least one isolate, with 2,668 of these sites being mutated in multiple isolates. The sites mutated in at least two isolates had a mean of 26.25 mutated isolates, and a median of 3. To identify the distribution of common mutations by the number of isolates they are found in, we identified the top 50 mutated sites and the number of isolates with a non-reference resolved base in those sites (Fig. 1). Three nucleotides, 3037, 14408 and 23403, were found to be mutated in over 7,000 isolates. Out of these three, 14408C>T was previously established as a mutation of interest for the RdRp gene. 23403A>G is a nonsynonymous mutation in the surface glycoprotein, while 3037C>T is a synonymous mutation in nsp3, a replication scaffolding protein (Yin, 2020). Notably, only 26 of the top 50 sites were found in the Orf1ab coding region, despite it comprising 71% of the SARS-CoV-2 genome, with the percentage being 57%, 61% and 65.3% when we consider top 100, top 200, and all mutated sites, respectively. Other than 14408C>T, two of the top 50 mutations were also in the RdRP coding sequence, although both of them are synonymous mutations that presumably

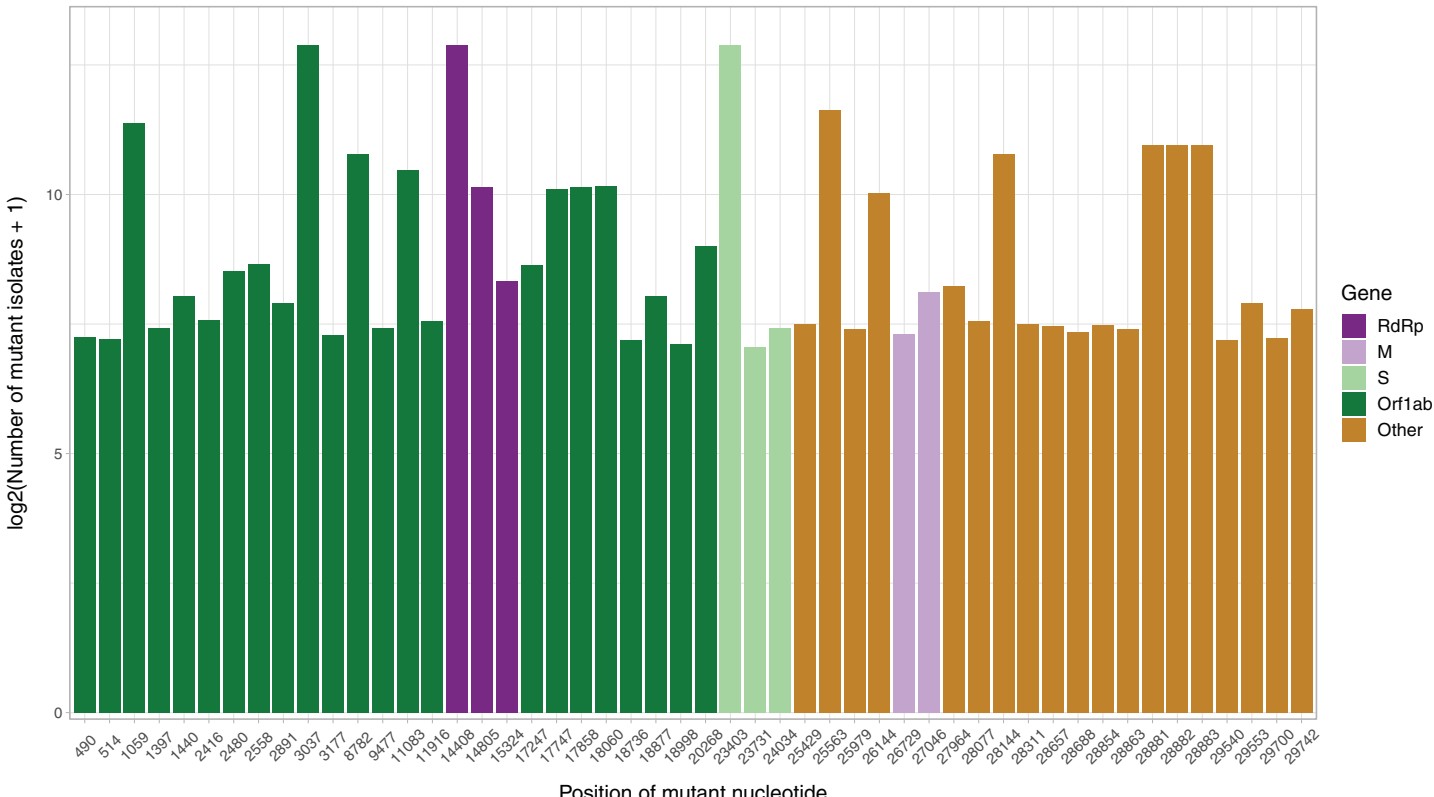

**Figure 1 Bar graph of top 50 most mutated nucleotides vs. log2-transformed number of samples with non-reference nucleotide at position.**
The *x*-axis represents the position of the nucleotide in the reference genome, the *y*-axis represents log 2 of number of isolates with disagreeing nucleotide aligning to the position in sequence plus 1. Unresolved sequence calls during library sequencing or gaps are not included in the number of isolates. Colors of the bars indicate the gene locus or mature peptide region where the nucleotide is in, with the RdRp mature peptide being considered separately from the remainder of the Orf1ab region. The 5′ untranslated region and the 3′-most 100 nucleotides are not included in the graph.

do not affect the protein structure. Two of the top 50 sites were found in the membrane glycoprotein coding region, with 26729T>C being a synonymous mutation and 27046C>T being a nonsynonymous mutation causing a T175M mutation in the peptide sequence. None of the top 50 mutated sites were found in the envelope glycoprotein region, which has only 23 sites mutated in multiple isolates.

We then examined the distribution of mutant RdRp, envelope, and membrane protein genes by geographical location (Africa, Asia, Europe, North America, Oceania and South America), in order to see what percentage of isolates from each region carried a mutation in these coding regions. South America had the highest percentage of RdRp mutants, 93.22%, while Asia had only 32.71%, the lowest among the regions. South America also had the highest number of mutant isolates for the M gene, and the second highest for the E gene, at 11.02% and 2.54% respectively (Fig. 2). However, it should be noted that Africa and South America both have remarkably lower number of sequences available and the disease showed a relatively late spread in these two continents compared to others.

The first case in South America (EPI_ISL_412964) was a 61 year-old male who travelled to Northern Italy and returned on February 21. This patient-zero carried the SARS-CoV-2

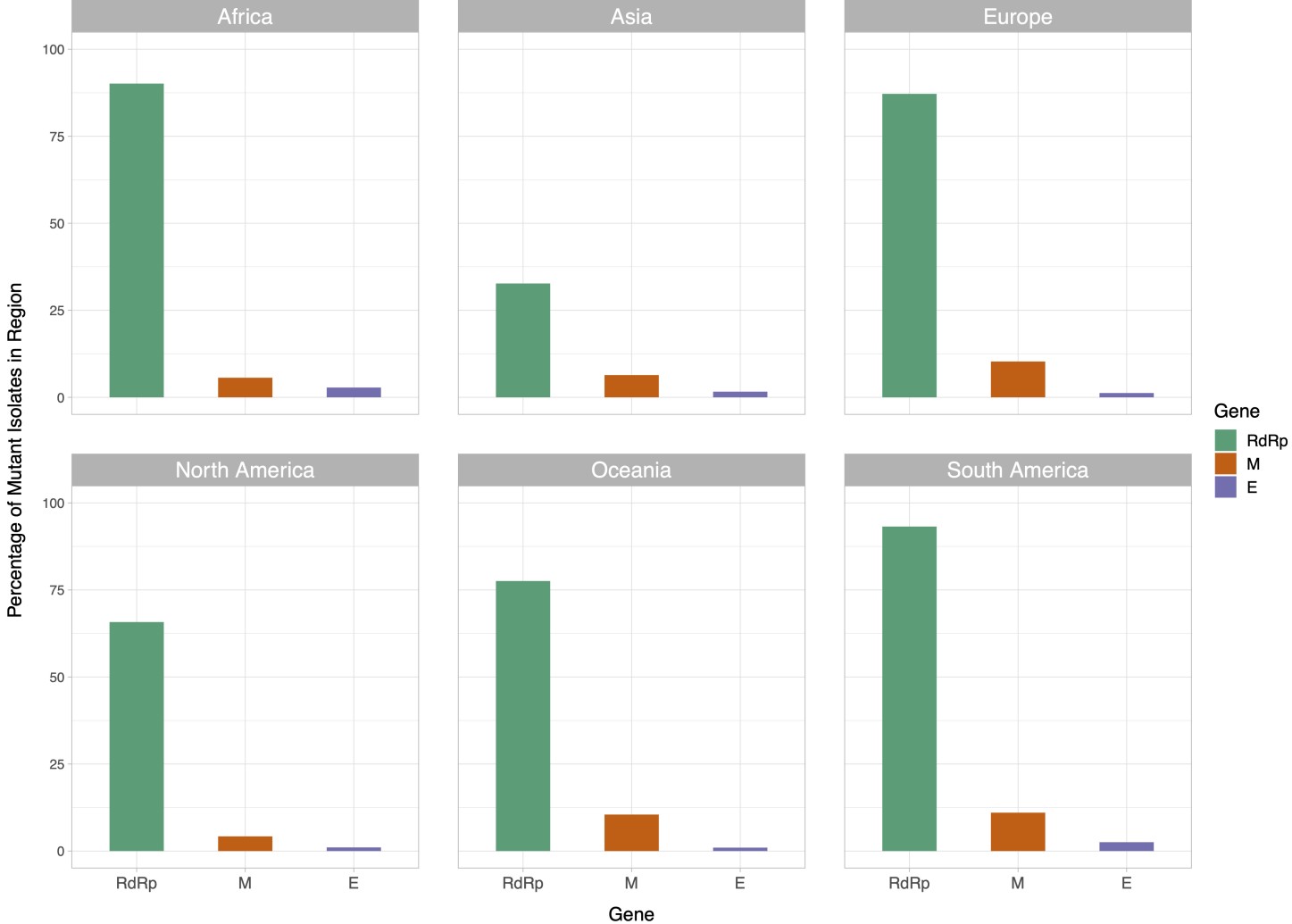

**Figure 2** Bar graph of percent of isolates per region containing non-reference coding sequences for the proteins envelope glycoprotein (E), membrane glycoprotein (M), and RdRp protein. Violet bars represent percent of isolates with mutant envelope glycoprotein, while orange bars represent the same percentage for membrane glycoprotein, and green bars represent the same for RdRp.

strain dominant in Italy and led to local transmission, which may explain the high-frequency of mutations at nucleotides 3037, 14408 and 23403 in Brazil. Interestingly, a few other cases who also returned from Northern Italy within a few days of patient-zero did not spread the disease as effectively. On the other hand, high-frequency of isolates with M or E mutations in South America can possibly be explained by relatively late spread of the disease ("GISAID Initiative", https://www.epicov.org/epi3/frontend#272e13).

## Associations between MoE and top ten frequently observed mutations in RdRp

To examine how the most common mutations in RdRp affect mutation rate of the SARS-CoV-2 genome, we identified the 10 most frequently mutated nucleotides in the RdRp region. Table 1 summarizes the frequencies and comparisons of MoE between RdRp

**Table 1 Comparisons of MoE and RdRp mutations.**

| Mutations | Values | MoE absent (*n*: 10,167) % (*n*) | MoE present (*n*: 1,041) % (*n*) | Total (*n*: 11,208) % (*n*) | *p* |
|---|---|---|---|---|---|
| 14408 | Absent | 38.4 (3,905) | 27.7 (288) | 37.4 (4,193) | <0.001* |
|  | Present | 61.6 (6,262) | 72.3 (753) | 62.6 (7,015) |  |
| 14805 | Absent | 89.6 (9,113) | 94.3 (982) | 90.1 (10,095) | <0.001* |
|  | Present | 10.4 (1,054) | 5.7 (59) | 9.9 (1,113) |  |
| 15324 | Absent | 97.0 (9,866) | 99.6 (1,037) | 97.3 (10,903) | <0.001* |
|  | Present | 3.0 (301) | 0.4 (4) | 2.7 (305) |  |
| 13730 | Absent | 99.1 (10,074) | 99.8 (1,039) | 99.2 (11,113) | 0.011* |
|  | Present | 0.9 (93) | 0.2 (2) | 0.8 (95) |  |
| 14786 | Absent | 99.2 (10,089) | 99.7 (1,038) | 99.3 (11,127) | 0.085 |
|  | Present | 0.8 (78) | 0.3 (3) | 0.7 (81) |  |
| 13536 | Absent | 99.5 (10,112) | 99.9 (1,040) | 99.5 (11,152) | 0.060 |
|  | Present | 0.5 (55) | 0.1 (1) | 0.5 (56) |  |
| 13862 | Absent | 99.6 (10,128) | 99.4 (1,035) | 99.6 (11,163) | 0.349 |
|  | Present | 0.4 (39) | 0.6 (6) | 0.4 (45) |  |
| 13627 | Absent | 99.6 (10,130) | 99.9 (1,040) | 99.7 (11,170) | 0.256 |
|  | Present | 0.4 (37) | 0.1 (1) | 0.3 (38) |  |
| 14877 | Absent | 99.6 (10,129) | 100.0 (1,041) | 99.7 (11,170) | – |
|  | Present | 0.4 (38) | – | 0.3 (38) |  |
| 15540 | Absent | 99.6 (10,130) | 99.9 (1,040) | 99.7 (11,170) | 0.256 |
|  | Present | 0.4 (37) | 0.1 (1) | 0.3 (38) |  |

**Note:**
 * *p*-value < 0.05 was statistically significant.

mutants. There are statistically significant associations between the mutations at nucleotides 13730, 14408, 14805, 15324 and MoE ($p < 0.05$). However, our statistical analysis indicates that there are no significant associations between the mutations 13536, 13627, 13862, 14786, 14877, 15540 and MoE ($p > 0.05$).

We also tested for possible associations between MoE and time (in days). Mean days in the MoE absent and present groups were 85.80 ± 15.56 and 85.77 ± 15.61, respectively. The days variable in the MoE absent group was not statistically significantly higher than the MoE present group (Median (IQR): 88.00 (15) vs 86.00 (18); $p = 0.095$). Because it was not statistically significant, we did not include "days" in the logistic regression models.

## Associations between MoE and geographic locations

Distribution of SARS-CoV-2 mutations show variability among geographical locations, mainly due to founder effects, as well as various other epidemiological factors. In order to compare the distribution of MoE among different geographic locations, Table 2 shows that there are statistically significant associations between the locations and MoE ($p < 0.001$). The most frequently observed location for the MoE is Europe ($n = 658$), however, it is largely due to higher representation of European viral genomes in the GISAID database.

**Table 2 Distribution of MoE across geographical locations.**

| Locations | Moe absent % (n) | Moe present % (n) | Total % (n) | p |
|---|---|---|---|---|
| Africa | 91.5 (65) | 8.5 (6) | 100.0 (71) | <0.001* |
| Asia | 92.0 (790) | 8.0 (69) | 100.0 (859) | |
| Europe | 88.6 (5,111) | 11.4 (658) | 100.0 (5,769) | |
| North America | 94.8 (3,194) | 5.2 (176) | 100.0 (3,370) | |
| Oceania | 88.5 (904) | 11.5 (117) | 100.0 (1,021) | |
| South America | 87.3 (103) | 12.7 (15) | 100.0 (118) | |
| Total | 90.7 (10,167) | 9.3 (1,041) | 100.0 (11,208) | |

**Note:**
* $p$-value < 0.05 was statistically significant.

**Table 3 Logistic regression model of MoE and location on single variables.** Each location was represented as itself (1) and others (0).

| Locations | p | OR | 95% CI |
|---|---|---|---|
| Africa | 0.807 | 0.901 | [0.389–2.084] |
| Asia | 0.188 | 0.843 | [0.653–1.087] |
| Europe | <0.001* | 1.700 | [1.490–1.939] |
| North America | <0.001* | 0.444 | [0.376–0.525] |
| Oceania | 0.012* | 1.297 | [1.058–1.591] |
| South America | 0.200 | 1.428 | [0.828–2.465] |

**Notes:**
OR, odds-ratio; CI, confidence interval.
* $p$-value < 0.05 was statistically significant.

The highest proportion of MoE is seen in South America (12.7%), whereas North America has the lowest (5.2%).

## Logistic regression models of the MoE

Next, we evaluated location in the univariate logistic regression models of the MoE (absent (0) and present (1)) for each location (itself (1) and others (0)) (Table 3). Europe, North America and Oceania were found statistically significant to predict MoE ($p < 0.05$). While the odds ratio for Europe was 1.700 (95% CI [1.490–1.939], $p < 0.001$), odds ratios for North America and the Oceania were 0.444 (95% CI [0.376–0.525]; $p < 0.001$) and 1.297 (95% CI [1.058–1.591]; $p = 0.012$) for the presence of MoE, respectively. Thus, our results suggest that SARS-CoV-2 genomes in Europe and Oceania are more likely to have MoE compared to other locations (1.7 and 1.3 times, respectively), while those in North America are >2.2 times less likely.

In the univariate logistic regression models of the MoE (absent (0) and present (1)), when the ten mutations were included separately in the models, 14408, 14805, 15324 and 13730 were found statistically significant to predict MoE ($p < 0.05$) (Table 4). In the final model (Final Model A), significant associations were also detected between MoE and these four mutations ($p < 0.05$). In the final model of the four mutations, the odds ratio for 14408 was 1.522 (95% CI [1.305–1.776]; $p < 0.001$) for the MoE. Thus, our results suggest

**Table 4 Logistic regression model of MoE on single variables and a final model.** (Final Model A) Model of four mutations on final model (Final Model B) Model of four mutations and location on final model.

| Mutations | Single variables | | | Final Model A | | | Final Model B | | |
|---|---|---|---|---|---|---|---|---|---|
| | p | OR | 95% CI | p | OR | 95% CI | p | OR | 95% CI |
| n14408 | <0.001* | 1.630 | [1.415–1.878] | <0.001* | 1.522 | [1.305–1.776] | 0.004* | 1.282 | [1.082–1.519] |
| n14805 | <0.001* | 0.519 | [0.396–0.681] | 0.008* | 0.673 | [0.502–0.903] | <0.001* | 0.478 | [0.352–0.648] |
| n15324 | <0.001* | 0.126 | [0.047–0.340] | <0.001* | 0.108 | [0.040–0.290] | <0.001* | 0.089 | [0.033–0.240] |
| n13730 | 0.028* | 0.209 | [0.051–0.847] | 0.049* | 0.243 | [0.060–0.992] | 0.025* | 0.201 | [0.049–0.820] |
| n14786 | 0.095 | 0.374 | [0.118–1.186] | – | – | – | – | – | – |
| n13536 | 0.086 | 0.177 | [0.024–1.279] | – | – | – | – | – | – |
| n13862 | 0.352 | 1.505 | [0.636–3.564] | – | – | – | – | – | – |
| n13627 | 0.188 | 0.263 | [0.036–1.921] | – | – | – | – | – | – |
| n14877 | 0.998 | 0.000 | [0.000 to –] | – | – | – | – | – | – |
| n15540 | 0.188 | 2.263 | [0.036–1.921] | – | – | – | – | – | – |
| Location | <0.001* | – | – | | | | <0.001* | | |
| Africa | 0.901 | 1.057 | [0.442–2.527] | – | – | – | 0.574 | 1.292 | [0.528–3.160] |
| South America | 0.092 | 1.667 | [0.920–3.023] | – | – | – | 0.263 | 1.416 | [0.770–2.605] |
| Europe | 0.003* | 1.474 | [1.138–1.910] | – | – | – | 0.036* | 1.353 | [1.020–1.794] |
| North America | 0.002* | 0.631 | [0.473–0.842] | – | – | – | <0.001* | 0.537 | [0.398–0.725] |
| Oceania | 0.014* | 1.482 | [1.084–2.025] | – | – | – | 0.026* | 1.448 | [1.045–2.008] |

Notes:
OR, odds-ratio; CI, confidence interval; Multiple logistic regression final model was executed on all these statistically significant variables, included together in the model, and selected with the backward stepwise method.
* $p$-value < 0.05 was statistically significant.

that SARS-CoV-2 genomes with the 14408C>T mutation are 1.5 times more likely to have MoE. We also evaluated "location" in the univariate logistic regression models of the MoE, and found that it was statistically significant to predict MoE ($p > 0.001$). Therefore, the final model of logistic regression analysis for independent variables 14408, 14805, 15324, 13730 and "location" (Asia is the reference group) was then built to evaluate their associations with MoE (Final Model B). This final analysis revealed that the same four mutations (14408, 14805, 15324, 13730) and "location" were significantly associated with MoE ($p < 0.05$). Europe, North America and Oceania were found to be statistically significant on single variable model for the location to predict MoE ($p = 0.003$, $p = 0.002$ and $p = 0.014$, respectively); similarly, the same locations were also statistically significant in the final model ($p = 0.036$, $p < 0.001$ and $p = 0.026$, respectively). According to the final model that was constructed with four mutations and location (Final Model B), viral genomes both in Europe and Oceania are more likely to have MoE compared to genomes in Asia (1.35 and 1.45 times, respectively). These results indicate that both RdRp mutations and location independently predict MoE status.

Whereas the 14408C>T mutation predicted higher risk of MoE, the other three significant mutations in RdRp predicted a lower risk, particularly the 15324C>T mutation, which predicted about 10-fold reduced risk of MoE. Although location was another predictor of MoE, as expected, multivariate logistic regression analysis indicated that the

association between RdRp mutations and mutational status of M or E genes was independent from location.

Two of the four significant RdRp mutations were first detected around the same time: 15324 on January 22 and 14408 on January 24 both in Asia; 14805 was first detected in a European genome on February 9, and the most recent of the four, 13730 was detected in an Asian genome on March 4. Despite arising within a few days' interval (based on the first genomes which they have been detected in so far), 14408 and 15324 display >20-fold difference in their spread: 14408 ($n$ = 7,015 genomes) vs 15324 ($n$ = 305 genomes). Although this observation may be explained by better adaptation of the viruses with a mutation that cause increased mutation rate to changing environments, it could as well be explained by founder effects, genetic drift, and other epidemiological factors. More data and particularly functional studies where mutant viruses can be compared side by side will be required to test this hypothesis.

Our observation that the two different mutants of RdRp result in ~14-fold difference in the likelihood of having mutations in parts of the genome that evolve relatively slow and are under less selective pressure (M and E genes) supports the hypothesis that mutations of RdRp contribute significantly to the SARS-CoV-2 genome evolution. A mutant RdRp that is more error-prone would be expected to increase viral genetic diversity and allow the virus to spread under different selective pressures, such as spreading to different populations. As lower-fidelity is also associated with higher speed, such mutations may allow higher titers of virus within host cells. On the other hand, a higher fidelity polymerase would be more suitable where optimal conditions are reached and errors in replication would be costly. Although preliminary studies suggest that 14408C>T (P323L in RdRp) could lower replication fidelity, it is less clear how the synonymous 15324C>T mutation could lead to lower mutation rates (*Begum et al., 2020*). It should be noted that 288 of 305 (94.4%) genomes worldwide with the 15324C>T mutation also have the 14408C>T mutation, and MoE rate is 1.39% (4/288) among double mutants, whereas it is 11.13% (749/6,727) for 14408C>T-only mutants. It is possible that 15324C is part of an as-yet-unknown viral sequence that interacts with host factor(s) and 15324C>T mutation indirectly affects the 14408C>T mutation through modulation of this interaction; as there are currently only 305 genome sequences available with this mutation, this question may be better answered as more viral genome sequences accumulate and functional studies are performed.

Three other mutations that co-evolved and are seen together with 14408C>T are 23403A>G (D614G in S protein) and 3037C>T (F106F in NSP3). The first 14408C>T mutation dates to a patient whose sample was collected on January 24 in China, but sequenced and submitted to GISAID on April 10. However, it took 27 days for the second case with the same mutation to appear, and interestingly, not in China, but in Italy. Two days later, on February 22, the first case with 14408C>T was reported in Australia, 31 days after the first SARS-CoV-2 case in the country. Following its introduction to Europe, it took another 10 days for the emergence of the second case in Asia, which can be possibly attributed to strict measures taken by the authorities that led to a steep decline in viral spread particularly in China. In contrast to Europe, North and South America, where

14408C>T became the dominant form together with its co-mutations (23403A>G and 3037C>T), 14408C>T and its co-mutations remained as the minor form in Asia, 14408C>T being present in only 15.9% (137/859) of viral genomes. Emergence of 14408C>T in South America, North America and Africa was 5, 7 and 8 days following the first European mutant virus, and again became the dominant form, as 81.3%, 59.4% and 80.3% of viral genomes carry the mutation, respectively.

A recent study postulated that one of the co-mutations of 14408C>T, namely 23403A>G that causes D614G mutation in the S protein, may result in a more transmissible form of SARS-CoV-2 (*Korber et al., 2020*). This claim was based mainly on the observation that D614G mutant virus became the dominant form in more than one geographical location upon its introduction, as summarized above for its co-mutation 14408C>T. However, in the absence of mechanistic insight to explain how this particular mutation could lead to higher transmissibility, other explanations based on stochastic factors are equally possible. A study by *Bhattacharyya et al. (2020)* suggested that D614G mutation creates an additional protease cleavage site near the S1–S2 junction, which may increase the success of viral integration with the host cell, and linked its dominance in Europe to certain human variants that control expression of TMPRSS2.

On the other hand, it is intriguing that between the first appearance of three co-mutations (on 14408, 23403, 3037) on January 24 in a Chinese isolate (EPI_ISL_422425) and their second co-appearance on February 20 in an Italian isolate (EPI_ISL_412973), there are at least six and possibly eight different viral genomes where three of the four co-mutations exist, with the exception of 14408C>T: on January 28 in Germany (EPI_ISL_406862), on February 5, 6, 7 and 8 in China (EPI_ISL_429080, EPI_ISL_429081, EPI_ISL_416334, EPI_ISL_412982, and EPI_ISL_429089); and two more Chinese viral sequences that failed our quality control standards and were eliminated from the overall analysis. Despite weeks of existence, 23403A>G became the dominant form only after the appearance of the first Italian case with all four mutations on February 20. If this form of SARS-CoV-2 is really more transmissible, the next question that needs to be answered is whether it is due to any one of the three mutations alone, or whether a combination of two or three are needed. Based on the lack of successful spreading of the virus in its absence and our results showing increased mutability in its presence, we speculate that 14408C>T could be cooperating with the other two mutations. Alternatively, altered mutation rate may be a byproduct and the RdRp mutations may act through speeding up or slowing down the replication process, which would in turn affect the viral load and virulence. Also, viruses with mutant RdRp may become more resistant to anti-viral drugs, such as the commonly used remdesivir. Such implications make RdRp mutations attractive targets for epidemiological and functional studies with direct therapeutic implications.

## CONCLUSIONS

Effects of different mutations on SARS-CoV-2 phenotypes (i.e., mutation rate, transmissibility, virulence, immune evasion etc.) are hot topics of research as there is an intense race worldwide to develop therapies and understand the viral biology. Some of these

studies suggested that RdRp and spike protein mutations could significantly affect the virus behavior and therefore the human health. Our study sheds light on the effects RdRp mutations, particularly 14408C>T mutation, on the mutability and possibly transmissibility of SARS-CoV-2. Further functional studies are required to test our findings.

## ACKNOWLEDGEMENTS

The authors would like to thank Mr. Alirıza Arıbaş and Mr. Ahmet Bursalı from Izmir Biomedicine and Genome Center for their technical assistance. The authors would also like to extend their thanks to the Izmir Biomedicine and Genome Center (IBG) COVID19 platform IBG-COVID19 for their support in implementing the study.

### Funding

Yavuz Oktay is supported by the Turkish Academy of Sciences Young Investigator Program (TÜBA-GEBİP). The funders had no role in study design, data collection and analysis, decision to publish, or preparation of the manuscript.

### Grant Disclosures

The following grant information was disclosed by the authors:
Turkish Academy of Sciences Young Investigator Program: TÜBA-GEBİP.

### Competing Interests

Aslı Suner and Gökhan Karakülah are Academic Editors for PeerJ.

### Author Contributions

- Doğa Eskier conceived and designed the experiments, performed the experiments, analyzed the data, prepared figures and/or tables, authored or reviewed drafts of the paper, and approved the final draft.
- Gökhan Karakülah conceived and designed the experiments, performed the experiments, analyzed the data, prepared figures and/or tables, authored or reviewed drafts of the paper, and approved the final draft.
- Aslı Suner conceived and designed the experiments, performed the experiments, analyzed the data, prepared figures and/or tables, authored or reviewed drafts of the paper, and approved the final draft.
- Yavuz Oktay conceived and designed the experiments, performed the experiments, analyzed the data, prepared figures and/or tables, authored or reviewed drafts of the paper, and approved the final draft.

### Data Availability

The data is available at Mendeley: Eskier, Doğa; Karakülah, Gökhan; Suner, Aslı; Oktay, Yavuz (2020), "SARS-CoV-2 GISAID isolates (2020-May-5) genotyping VCF", Mendeley Data, v1 DOI 10.17632/x4t94w9njt.1.

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
