# Peer review of "RdRp mutations are associated with SARS-CoV-2 genome evolution"

_PeerJ, doi:10.7717/peerj.9587_

## Round 0.1 · original submission · Minor Revisions

Dear Dr. Oktay,

Please follow the required changes of both reviewers and provide a revised version following all the details observed by both reviewers. Reviewer #1 provided a very detailed analysis and suggests changes in the main text structure and result presentation. Please follow all the suggested changes, as well as the detailed analysis of typos and sentence structure. Reviewer #2 raises two important issues one regarding the differences in mutations among viruses from South America region and the remaining of the world. The reviewer also suggests a better association and description of genetic variations in SARS-CoV-2 and “epidemiological factors”. Overall these are largely minor corrections, but should be carefully performed by the authors.

Reviewer 1 ·

Basic reporting

The manuscript is mostly well written and referenced. I have some suggestions of modifications to be made on text, tables and figure presentations (explained below, at the general comments for the author)

Experimental design

The research theme is very actual, relevant and interesting. The experimental design seems rigorous enough and is also described with sufficient details for understanding and replication.

Validity of the findings

It seems to me that the authors performed an honest and robust analysis and that their fundamental conclusions are consistently based on the generated results.

Additional comments

I consider that the manuscript entitled “RdRp mutations are associated with SARS-CoV-2 genome evolution” contains important and relevant new information, which will have impact on the understating of SRS-CoV-2 worldwide spreading, evolving and adapting. The authors made good and extensive use of public available sequence data on SARS-CoV-2, constructed an interesting hypothesis on the relationship between occurrence of RdRp mutations and overall virus mutation rate and also performed a consistent and robust statistical analysis in order test it.
I have some observations regarding main text structure and results presentation which, I believe, will collaborate to improve the readability of the manuscript.

1. As, by the final of results/conclusion section, the authors make interesting considerations about emergence and spreading of SARS-Cov-2 mutations, I recommend the inclusion of information regarding time and origin of the reference viral genome sequence (NC_045512.2), as it is important as a referential point to this analysis.
2. I believe it would be very interesting and visually informative if the authors could transform the textual time/geographic information about SARS-Cov-2 mutations seen on the final paragraphs in a third figure, containing a map of the globe, dates and supposed directions of spreading between continents. It is just a suggestion, but I think it would make a nice ending to the manuscript.
3. Figure 1. The information regarding gene names and colors is already present below the X axis. So, it is not necessary at all for it to appear in a legend.
4. Tables 1 and 2 have very important data, but it is somewhat difficult to discern the relationship between presented values, as there are a lot of columns and numbers. As it seems that the percentage values are more important to interpret the data, I suggest the authors to adjust the presentation of theses tables by displaying values of percentage with sample size within parentheses.
Thus, data in the first row would appear this way:
38.4 (3905) 27.7 (2888) 37.4 (4193)
This way it will be easier to the reader to compare the values and understand the table.

Next, I present a list of comments, remarks and suggestions that, I hope, will improve the understanding of the text.

Row 65: You probably means " (...) not yet fully known".

Row 87: The end of this sentence is confusing. I think it means "(...) multiple forms of virus, varying on their mutation rate across the globe", or something similar.

Row 91: Consider changing "a" for "with".

Row 96: Consider changing sentence to "(...) under locus ID NC_045512.2".

Row 98: Consider changing sentence for " (...) which are not A, C, G, T, N, or -)".

Row 140: Please replace comma (,) for an end (.).

Row 144: Although it appears later in the text, it would be good to briefly explain here why this mutation is of interest.

Row 151: In order to improve the reading, I suggest to change this sentence to "Two of the top 50 sites were found (...)".

Rows 167-168 and 169: I suggest to list the nucleotides in crescent order.

Rows 190 to 192: The structure of this sentence is somewhat strange. Please, consider to revise it.

Rows 208 - 210: The same as above. The structure of this sentence is also somewhat strange. Please, consider to revise it.

Row 236: Please insert reference for these studies.

Row 240: Seems to me improbable (but not impossible) that a single nucleotide position will modulate the interaction of viral genome with host factors. It would be more adequate to say that "It is possible that 15324C may be part of a DNA motif that modulates the interaction of viral genome and host factors", or something similar.

Rows 264 to 266: Please consider to revise sentence construction. I think it would be better if it reads "However, unless mechanistic insight arises that could explain how this particular mutation could lead to higher transmissibility, other explanations based on stochastic factors are equally possible.”.

Row 266: Insert "et al."

Row 272: Please insert the reference/code for the Italian isolate, as shown above for the Chinese one.

Row 277 - 278: The segment "(...), unless completely confined and eliminated," sounds strange and not related to the whole sentence. Please revise it to improve the understanding.

Row 286: Consider changing it to "Also, mutant RdRp may turn the virus more resistant to anti-viral drugs, such as the commonly used remdesivir.", as drug resistance is relative to the virus, not the polymerase.

Row 384: There is unnecessary redundancy of the world protein in this sentence. Consider changing it to “Bar graph of percent of isolates per region containing non-reference coding sequences for the envelope glycoprotein (E), membrane glycoprotein (M), and RdRp protein.


Finally, hoping that the authors will carefully consider my recommendations, I consider that the manuscript should be accepted for publication by Peer Journal with minor revisions and without need to re-review it. In the urgent context of Covid-19 pandemics, I think it is important that this information can be available to scientific community as soon as possible.

·

Basic reporting

The authors describe the importance of the mutations found in RNA-dependent RNA polymerase (RdRp) in relation to the genetic variation found in SARS-CoV-2. These are related to the evolution of virus variants according to mutation rates. The authors are also related to mutation rates found with regions of the world where the genome has been sequenced. The description of this molecular mechanism is essential to understand how SARS-CoV-2 is spreading around the world and opening perspectives on possible drug developments used. The professional language in English is consistent, as are the results presented through graphs and tables. The hypothesis discussed is in accordance with the data presented and with the literature provided in the manuscript.
Minor revision should be paid attention to the formatting of references, as some references that do not have the DOI number and in italic letter, are not all in the same pattern according to the journal. Throughout the text there are also grammatical errors in punctuation.

Experimental design

The research data is within the scope of the paper and defined in the materials and methods. Despite the research supplanting and adding interesting data in research related to SARS-CoV-2, with methodologies that can be replicated, there are some doubts. In figure 2, related to the percentage of isolates by region in relation to the studied targets and, according to table 2, the South America region has the highest percentage of mutants in RdRp, but have the lowest number of sequenced genomes. The authors should discuss more about this result, in which it remains valid, but qualitatively different from other regions. In addition, include in the materials and methods the number of sequenced genomes of each region, facilitating the reading and interpretation of the results.

Validity of the findings

There is no need to repeat the experiments, being well analyzed according to the existing literature. But, in some parts of the text, the actors associate genetic variations in SARS-CoV-2 with “epidemiological factors”, but do not quote and describe them. It is important to include these factors in the text, even if they are not addressed experimentally. The conclusions are in line with the discussion and linked to the central research of the work.

Additional comments

The work sent to PeerJ - the Journal of Life and Environmental Sciences addresses in a solid way the relationship of mutations in RdRp with the evolution of the SARS-CoV-2 genome. Despite a scarce literature, it is noted the importance of genomic studies and their aspects and relationships with the dispersion throughout the world of the COVID-19 virus. The text, despite small revisions to be made, manages to associate, discuss, and associate the analyzed results.

---

## Round 0.2 · accepted · Accept

Dear Dr. Oktay,

Congratulations and thanks for your detailed revision.